



# A new local meteoric water line for Inuvik (NT, Canada)

Michael Fritz[1], Sebastian Wetterich[1], Joel McAlister[2], Hanno Meyer[3]

[1]Permafrost Research, Alfred Wegener Institute Helmholtz Centre for Polar and Marine Research, Potsdam, Germany
[2]Aurora Research Institute, Aurora College, Inuvik, NT, Canada
[3]Polar Terrestrial Environmental Systems, Alfred Wegener Institute Helmholtz Centre for Polar and Marine Research, Potsdam, Germany

*Correspondence to*: Michael Fritz (michael.fritz@awi.de) and Sebastian Wetterich (sebastian.wetterich@awi.de)

**Abstract.** The paper presents a new local meteoric water line (LMWL) of precipitation stable oxygen and hydrogen isotopes from Inuvik in the Western Canadian Arctic. Data were obtained over 37 months between August 2015 and August 2018
resulting in 134 measurements of the isotopic composition of both types of precipitation, snow and rain. For 33 months of the sampling period each month is represented at least two times from different years. The new LMWL from Inuvik is characterized by a slope of 7.39 and an intercept of –6.70, and fills a data gap in the Western Arctic where isotopic composition data of precipitation are scarce and stem predominantly from before the year 1990. Regional studies of meteorology, hydrology, environmental geochemistry and paleoclimate will likely benefit from the new Inuvik LMWL. Data are available
on the PANGAEA repository under https://doi.pangaea.de/10.1594/PANGAEA.935027 (Fritz et al., 2021).

Keywords: LMWL, meteoric water line, GMWL, stable water isotopes, $\delta^{18}O$, $\delta^{2}H$, Deuterium excess

## 1        Introduction

The global annual average relationship between hydrogen ($\delta^{2}H$ or $\delta D$) and oxygen ($\delta^{18}O$) isotope ratios in natural meteoric waters is captured by the Global Meteoric Water Line (GMWL) that was developed by assembling numerous local annual mean records worldwide (Craig, 1961), and is expressed by the equation (1):

(1)        $\delta^{2}H = 8*\delta^{18}O + 10‰$

The GMWL is widely used as baseline in environmental geochemistry, meteorology, hydrology and hydrogeology to trace the
water cycle before and after recharge, and to use isotopic fractionation processes on the water pathway to infer source characteristics and transport mechanisms. The relationship of $\delta^{18}O$ and $\delta^{2}H$ in meteoric waters is controlled by mass-dependent fractionation of oxygen and hydrogen isotopes between evaporation from ocean seawater and condensation from vapour, which strongly depends on altitude, latitude, seasonality and continentality of a given location (e.g. Clark, 2013; Dansgaard, 1964; Gat, 1996).



A meteoric water line calculated for a given area is named as local meteoric water line (LMWL) with emphasis on the spatial
      variability in isotopic compositions of meteoric waters as investigated by Rozanski et al. (1993) by comparing the $\delta^{18}$O-$\delta^2$H
      relationships of monthly-scale samples at selected sites of the Global Network of Isotopes in Precipitation (GNIP)
      (IAEA/WMO, 2020). Thus, a LMWL represents the covariation of oxygen and hydrogen stable isotope ratios for a distinct
      area and observation period, and have practical utility as a hydrologic framework and benchmark for evaluating hydroclimatic
processes such as in isotope-enabled climate models (e.g. Werner et al., 2016). As a simplified, intuitive and site-specific
      representation of the average isotopic relationship in meteoric waters, coupled with information about the seasonal range of
      isotope composition in precipitation, a LMWL offers a reference for interpreting the isotope ratios measured in terrestrial
      waters and ice.

      A LMWL can deviate from the GMWL both in slope and intercept of the linear regression in a $\delta^{18}$O-$\delta^2$H co-isotope plot;
largely resulting from differences in humidity (e.g. Putman et al., 2019) that is tackled by the second-order parameter
      Deuterium excess ($d$; Dansgaard, 1964) and expressed by the equation (2):

      (2)      $d = \delta^2$H $- 8*\delta^{18}$O

      Deuterium excess is an indicator for non-equilibrium fractionation processes that might occur during phase transitions of water.
It has been established as a function of relative humidity in the moisture source regions. The statistical linear relationship of
      $\delta^2$H to $\delta^{18}$O varies both temporally and spatially, and the variation in the slope may hold information about seasonal
      climatology of the site (Craig, 1961; Rozanski et al., 1993).

      A correlation between air temperature (T) and precipitation $\delta^{18}$O was recognized and can be used to study surface air
      temperature change over time (Merlivat and Jouzel, 1979) using preserved ancient meteoric water such as in glacial ice (e.g.
Dansgaard, 1964; Jouzel et al., 2003) or in permafrost-bound ground ice (e.g. Meyer et al., 2015; Opel et al., 2018; Wetterich
      et al., 2021) to reconstruct paleoclimate. Such studies benefit from known modern hydroclimatic conditions as expressed in
      LMWLs (e.g. Porter et al., 2019; Porter and Opel, 2020).

      Although increasingly used, LMWLs from remote areas are rare, especially in the Arctic, as it requires long-term and regular
      sampling, analysing, and data processing (Putman et al., 2019). For example, in the Western Arctic where extensive
paleoclimatic research on (buried) glacial ice (e.g. Lacelle et al., 2007; Fritz et al., 2011; Coulombe et al., 2019), permafrost
      ground ice (e.g. Meyer et al., 2010; Fritz et al., 2011, 2012; St-Jean et al., 2011; Holland et al., 2020) and permafrost hydrology
      (e.g. Utting et al., 2012, 2013; Kokelj et al., 2015) takes place, the respective reference LMWLs to compare isotopic records
      with are based on scarce data covering only small periods of data acquisition before the year 1990 at only a few locations (Fig.
      1). The present data collection aims (1) to update and extend the previous Inuvik LMWL and, thus, (2) to improve the regional
framework for meteorological and hydrological applications of the modern environment as well as for paleoclimate studies in
      the region based on ancient ice stable isotopes.

## 2        Material and Methods

Inuvik (68.3°N, 133.5°W; Fig. 1) is located at the east channel of the Mackenzie River delta in the Northwest Territories (NT)
in Canada. The current WMO station (number: 719570) in Inuvik is located at the international airport at an elevation of 68
metres above sea level.

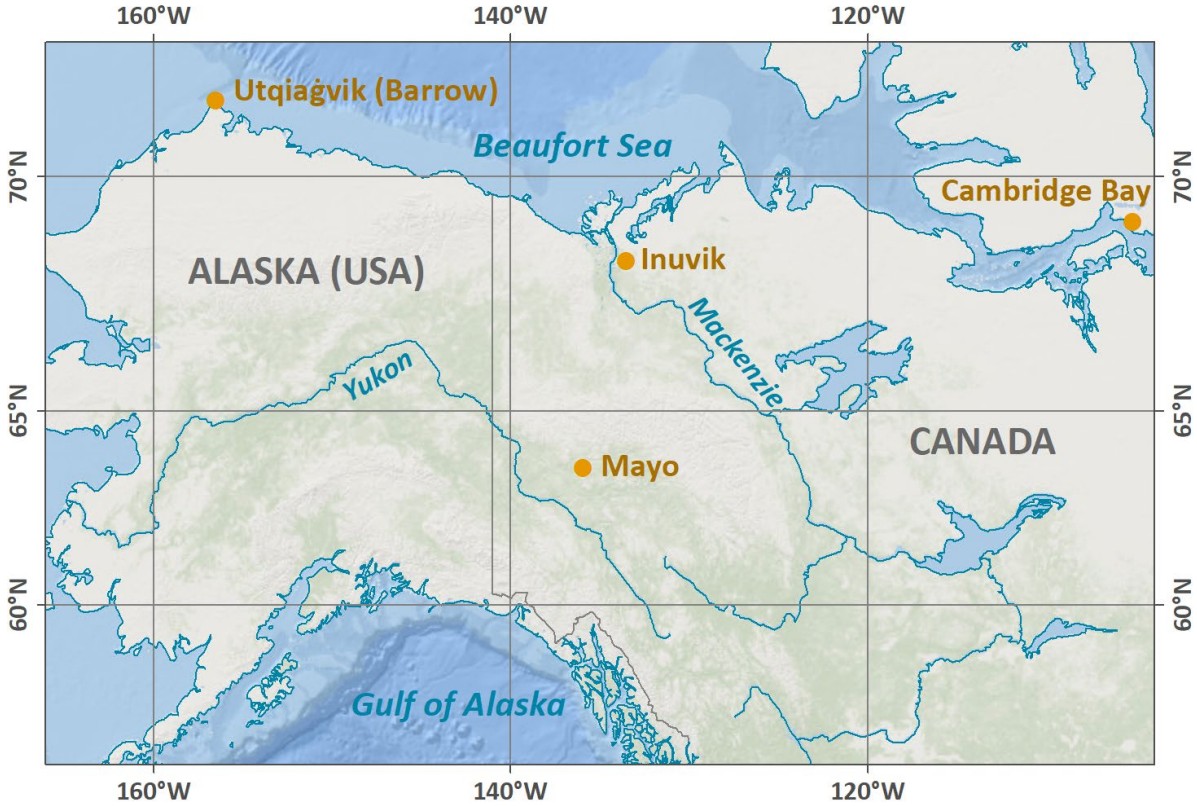

**Figure 1. Map with WMO stations in the western Arctic, where GNIP data and LMWLs exist (i.e., Inuvik, Utqiaġvik [Barrow], Mayo and Cambridge Bay). Map by Sebastian Laboor (AWI).**


The regular sampling of meteoric water for subsequent stable isotope analysis was conducted between August 2015 and August
2018 at the Aurora Research Institute (Western Arctic Research Centre, 191 Mackenzie Road, Inuvik, NT X0E 0T0, Canada)
in the town of Inuvik.

A precipitation gauge was constructed as a PVC funnel with 30cm diameter on top and fitting into a vertical PVC pipe with a
diameter of 10cm. A 250ml HDPE bottle is located behind the sleeve in the vertical PVC tubes, directly under the funnel,
which collects the water. The funnel and a 250ml collection bottle fit tightly together to avoid secondary evaporation. The
collection bottle can be taken out or replaced through the sleeve. Threaded rods in the inside of the PVC pipes run into the
ground for stabilization. Ropes and stakes stabilize the rain gauge against wind.





The rain gauge collection bottle was regularly checked and carefully emptied after every rain or snowfall event. Sampled
precipitation was transferred into 15ml or 30ml narrow-neck LDPE bottles, depending on the amount of water with no
remaining air in the bottle. Samples were stored dark and cool at the ARI in Inuvik at +4°C prior to analysis.

The oxygen ($\delta^{18}$O) and hydrogen ($\delta^{2}$H) stable isotope compositions of precipitation were measured in the Stable Isotope
Facility of the Alfred Wegener Institute in Potsdam (Germany), using a Finnigan MAT Delta-S mass spectrometer ($1\sigma < 0.1$‰
for $\delta^{18}$O, $1\sigma < 0.8$‰ for $\delta^{2}$H; Meyer et al., 2000). Values are given as per mil (‰) difference from the Vienna Standard Mean
Ocean Water (VSMOW) standard. A total of 135 samples were obtained and analysed from the monitoring period between
August 2015 und August 2018 of which 134 samples were considered in this study. One sample from February 2016 is
excluded from the data set and from interpretation due to its highly unusual isotopic composition ($\delta^{18}$O of –21.04‰, $\delta^{2}$H of –
186.1‰, $d$ of 17.8‰).

## 3        Data structure

The total of 134 valid datapoints represents 33 months of the entire monitoring period of 37 months (Table 1). The four months
without data during the observation period are February and May in 2016, and November and December in 2017.
Consequently, each month in the data set is represented at least two times from different years. Eight months of the year(s) are
represented at least three times from different years. The sample number per month varies between zero and up to 14 samples
(e.g. August 2018; Table 1).


**Table 1. Summary of sample representation across months and years.**

| Month | Year | 2015 | 2016 | 2017 | 2018 | Monthly sum |
|---|---|---|---|---|---|
| 1 | - | 1 | 3 | 1 | **5** |
| 2 | - | 0 | 5 | 2 | **7** |
| 3 | - | 1 | 2 | 1 | **4** |
| 4 | - | 5 | 1 | 2 | **8** |
| 5 | - | 0 | 2 | 6 | **8** |
| 6 | - | 3 | 2 | 3 | **8** |
| 7 | - | 4 | 3 | 7 | **14** |
| 8 | 4 | 11 | 8 | 14 | **37** |
| 9 | 12 | 6 | 7 | - | **25** |
| 10 | 4 | 1 | 1 | - | **6** |
| 11 | 2 | 4 | 0 | - | **6** |
| 12 | 2 | 4 | 0 | - | **6** |
| **Yearly sum** | **24** | **40** | **34** | **36** | |





The currently available GNIP data set for Inuvik precipitation originates from the 1980s. Data acquisition took place between 1986 and 1989. From the 48 months of these years only 14 paired ($\delta^{18}$O and $\delta^2$H) data points of monthly mean values exist.

The month of June is not represented in the previous Inuvik LMWL data set at all. All remaining months are represented at least one time, while December, January and February are represented twice.

The GNIP data (IAEA/WMO, 2020) provides monthly means of stable isotopes in precipitation. In order to keep comparability, we also present monthly means of the Inuvik LMWL although the entire database is freely available at PANGAEA (Fritz et al., 2021).

## 105    4      Data analysis

### 4.1    *Climate and stable isotope data during the observation period*

The meteorological monitoring data for the observation period are available from WMO station (number: 719570) in Inuvik (Environment and Climate Change Canada, 2021a; Fig. 2). The long-term average annual air temperature is –8.2°C and the annual air temperature amplitude amounts to 41°C with coldest mean temperatures in January and warmest mean temperatures

in July (1981-2010; Environment and Climate Change Canada, 2021b). The mean annual precipitation sums up to 241mm and snow depth reaches up to 159cm. During the observation period, the average annual air temperature was –5.5°C with an increasing trend towards today (Fig. 2). The maximum temperature amplitude between 2015 and 2018 was 58°C. The stable isotope composition of meteoric water expressed as monthly means varies over about 15‰ between –29.0 and –13.9‰ in $\delta^{18}$O (mean of –20.5‰), and over about 124‰ between –221 and –97‰ in $\delta^2$H (mean of –158‰). Deuterium excess ranges

from –2.4 to 14.4‰ (mean of 5.7‰, Table 2). The seasonal pattern in air temperatures is largely delineated by the stable isotope composition of precipitation with lower $\delta^{18}$O and $\delta^2$H values representing lower air temperatures (Fig. 2).





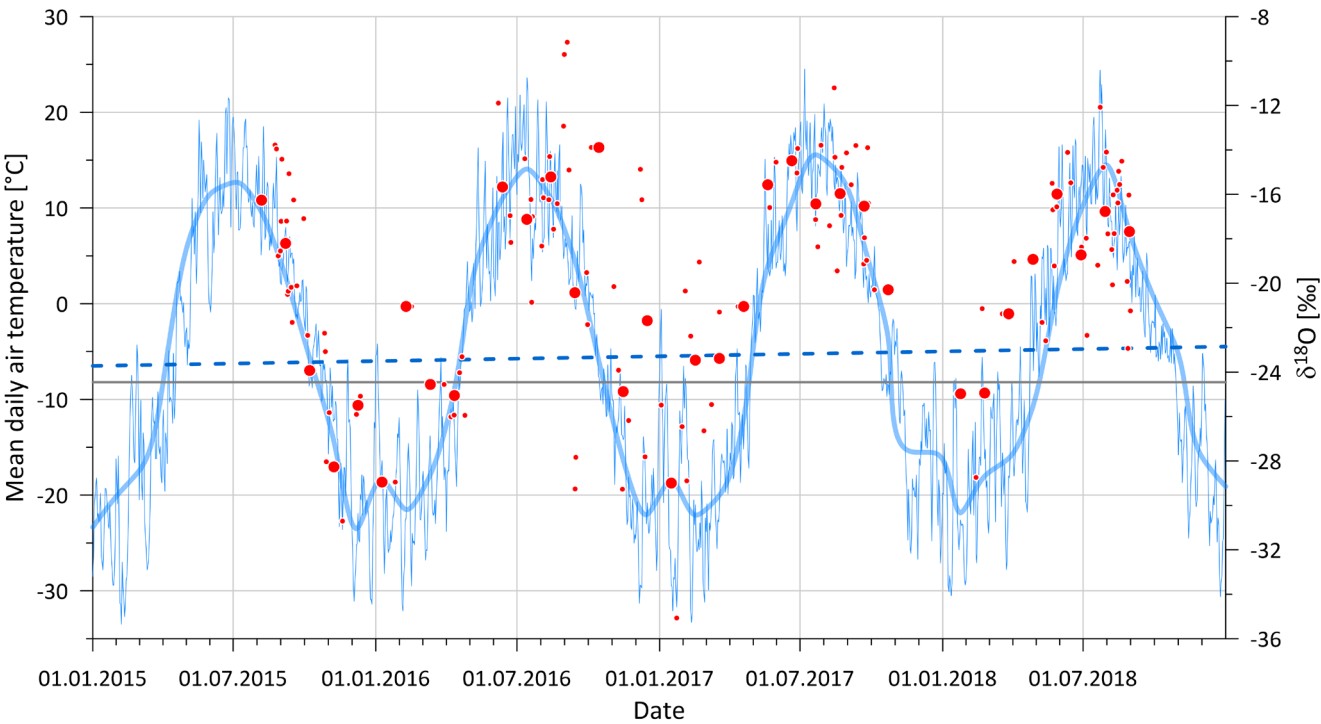

**Figure 2.** Air temperatures from Inuvik (NT, Canada) together with δ¹⁸O values in precipitation during the sampling period from
**August 2015 to August 2018. Thin blue line shows mean daily air temperatures and the thick blue line shows monthly averages.
Small red dots show all δ¹⁸O data points. Large red dots show δ¹⁸O monthly averages. Grey line illustrates long-term average of air
temperatures (1981-2010). Blue dashed line illustrates air temperature trend from 2015 to 2018.**

**Table 2. Basic statistics of stable water isotope data based on monthly means.**

|  | $\delta^{18}O$ | $\delta^2H$ | $d$ |
|---|---|---|---|
|  | [‰] | [‰] | [‰] |
| **MEAN** | –20.46 | –158.0 | 5.7 |
| **MEDIAN** | –20.30 | –155.3 | 5.8 |
| **MIN** | –29.00 | –221.1 | -2.4 |
| **MAX** | –13.89 | –96.7 | 14.4 |
| **25%-quantile** | –24.56 | –184.6 | 3.4 |
| **75%-quantile** | –16.43 | –127.8 | 8.4 |
| n | 33 | 33 | 33 |


*4.2       The new Local Meteoric Water Line from Inuvik*





The LMWL representing Inuvik precipitation between August 2015 and August 2018 is shown in Figure 3a in comparison to the previous LMWL covering 1986 to 1989 (Fig. 3b). Further LMWLs from the Western Arctic that are Utqiaġvik (Barrow),

Mayo and Cambridge Bay are summarized in Table 3. The slope of the new Inuvik LMWL is 7.39 and the intercept is –6.70. In comparison, the previous Inuvik LMWL shows values of 7.33 for the slope and –3.55 for the intercept (Fig. 3) underlining the long-term local singularity of stable isotope composition in meteoric water. The relationship between $\delta^2$H and $d$ in the Inuvik precipitation isotopic composition is expressed by equation (3):

(3)      $d = -0.07 * \delta^2\text{H} - 4.92$

with a moderate correlation of $R^2 = 0.47$ (Fig. 4). In general, low $\delta^2$H values, which are largely explained by low air temperatures, are associated with higher $d$ values. The mean annual $d$ for Inuvik is 5.7‰ compared to the global average of 10‰. The greatest difference between the new Inuvik LMWL and the previous Inuvik LMWL is associated with $d$ values. The dataset from between 1986 and 1989 contains only positive $d$ values that range between 3.6 and 34.4, with a mean of 14.9. In contrast, our $d$ data from between 2015 and 2018 contain a considerable number of negative values (Fig. 4), and they have a

smaller range and a much lower average (Table 2).

Other regional LMWL statistics differ distinctly from Inuvik (Table 3) emphasizing regional peculiarities.





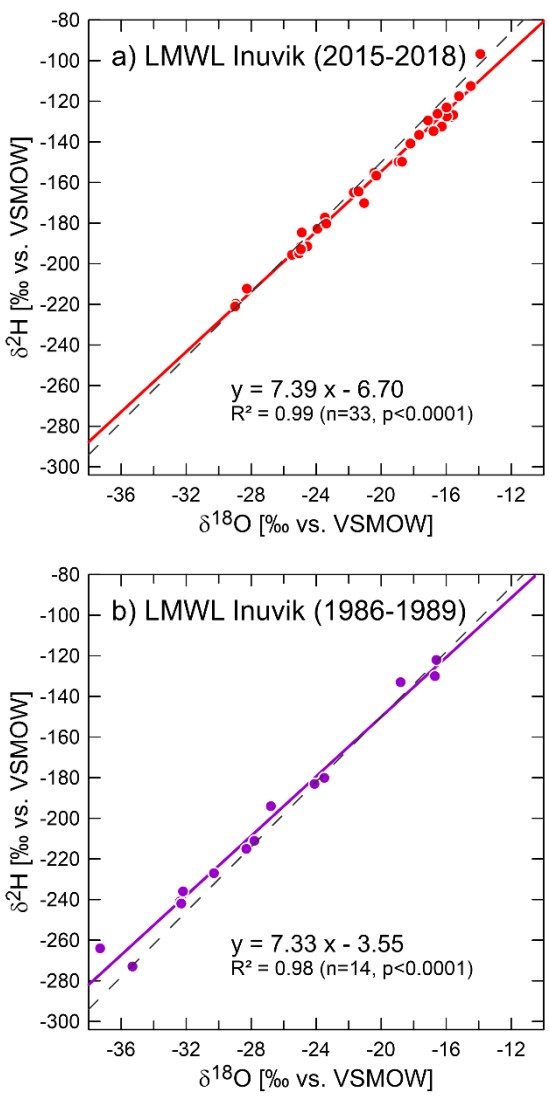

**Figure 3. Co-isotope plots of δ¹⁸O and δ²H of precipitation for Inuvik (NT, Canada). (a) New (2015-2018) local meteoric water line.**
**(b) Old (1986-1989) local meteoric water line. Dashed line represents the global meteoric water line.**

**Table 3. Available monthly-based LMWLs from the Western Arctic.**

| Location | WMO number | Coordinates | Period of data acquisition | Number of data points | LMWL equation |
|---|---|---|---|---|---|
| Inuvik - New | 719570 | 68.3°N, 133.5°W | 2015-2018 | 33 | $\delta D = 7.39 * \delta^{18}O - 6.70$ |
| Inuvik - Old | 719570 | 68.3°N, 133.5°W | 1986-1989 | 14 | $\delta D = 7.33 * \delta^{18}O - 3.55$ |
| Utqiaġvik (Barrow) | 943040 | 71.4°N, 156.5°W | 1962-1969 | 47 | $\delta D = 7.12 * \delta^{18}O - 9.06$ |
| Mayo | 719650 | 63.6°N, 135.9°W | 1985-1989 | 37 | $\delta D = 6.27 * \delta^{18}O - 36.86$ |
| Cambridge Bay | 719250 | 69.1°N, 105.1°W | 1989-1993 | 58 | $\delta D = 7.66 * \delta^{18}O + 0.83$ |




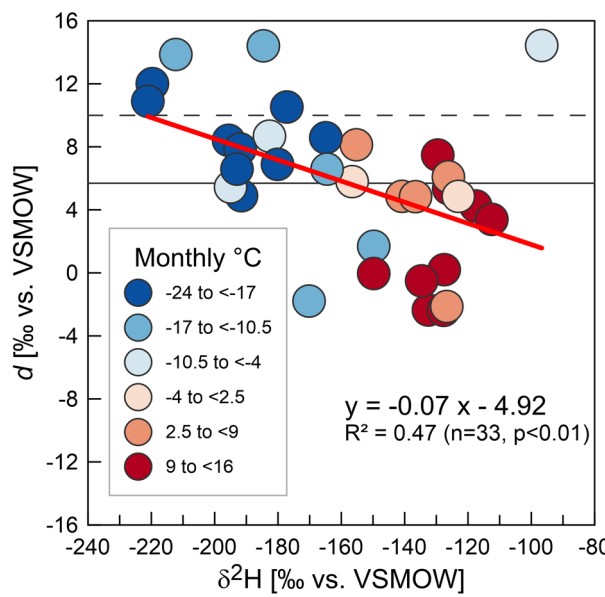

**Figure 4. Relationship between hydrogen isotope ratios ($\delta^2$H) and Deuterium excess *d* (monthly means). The solid horizontal line indicates mean *d* from Inuvik (this study), whereas the dashed line indicates global average *d*. Bubble colour indicates air temperature ranges (monthly mean) at the time of sampling. Red line represents the linear trend.**


The slight deviation of the new Inuvik LMWL slope if compared to that of the GMWL might either reflect regionally specific isotopic fractionation in equilibrium of distinct ranges in temperature and/or isotopic composition or point to seasonal precipitation that is characterized by non-equilibrium processes (Putman et al., 2019). The latter might affect the new Inuvik LMWL as snow from mixed-phase cloud processes controlled by Rayleigh distillation is likely to contribute to the annual

Inuvik precipitation, and is represented by high *d* values in cold-season precipitation (Fig. 4) and a slope below the GMWL slope of 8 (Fig. 3). The wide ranges over about 15‰ in $\delta^{18}$O and about 124‰ in $\delta^2$H of monthly means of the new Inuvik data set enable a rather well-defined LMWL. Putman et al. (2019) applied data quality criteria for LMWL calculations requiring time series that include at least three samples in each 3-month meteorological season to represent seasonality, while the length of the observation period depends on the timescale of the process of interest. In this context, the new Inuvik LMWL is

considered as valid representation of the isotopic composition of regional precipitation.

**Data availability**

Original data are available on PANGAEA under https://doi.pangaea.de/10.1594/PANGAEA.935027 (Fritz et al., 2021).



**Author contributions**

MF designed the study and analyzed the data. HM conceptualized the sampling device and analyzed the samples at the Stable
Isotope Facility at the AWI. MF and SW wrote the manuscript with contributions from all co-authors.

**Acknowledgements**

We greatly acknowledge the support of the Aurora Research Institute (ARI, Inuvik) for collecting the samples. AWI staff
members Hugues Lantuit, George Tanski, Jennifer Krutzke, Niklaas Schmidt, Mikaela Weiner, and Günther (Molo) Stoof are
thanked for technical support in the field and in the laboratory. This publication is part of the NUNATARYUK project that
has received funding from the European Union's Horizon 2020 research and innovation programme under grant agreement
No. 773421.

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
