# Peer review of "A new local meteoric water line for Inuvik (NT, Canada)"

_Earth System Science Data, 2021_

## Referee Comment (RC2)

**General comments**

The preprint is exceptionally well written. Very good English and concise wording. The whole preprint reflects that the authors have thorough knowledge of the given field of science. Plus, I could not identify any mistakes or faults in the referencing, which is rare.

**Specific comments/questions**

The authors goal is to provide an up-to-date and more robust dataset for calculating the local meteoric water line (LMWL) at Inuvik, Canada. To accomplish this goal, they collected event based precipitation samples from August 2015 to August 2018. The samples were stored in LDPE (low-density polyethylene) bottles at 4 °C. At this point I have a concern: the authors don't mention how long samples were stored before analysis. Samples were stored in Canada and the analyses were made in Germany, far away from each other, so it is reasonable to suppose that the samples were stored for several months. The International Atomic Energy Agency recommends HDPE (high-density polyethylene) bottles for storing water samples for several months, because LDPE bottles are not reliable. This is a very important question to make sure that the stored water samples have not suffered evaporation effect, because there is a small difference between the old and the new dataset and the most significant difference is in the d-excess. The average d-excess of the new dataset is significantly lower (5.7‰) than that of the old dataset (14.9‰). This difference can be caused by a) evaporation during sample storage, or b) climate change. Therefore, it is of paramount importance to check the reliability of sample storage.

I recommend authors to discuss this question in details. If they cannot rule out the possibility of the evaporation effect during the sample storage, then their results are very ambiguous. If they can, then the manuscript is practically ready for publication. I recommend to check the sample storage related to the old dataset as well.

**Mostly technical corrections**

Abbreviations (WMO in L65; AWI in Caption of Fig. 1; ARI in L81) should be defined at the first instance.

L88: The d value is -17.8‰, and not 17.8‰. Anyway, this very negative d value indicates evaporation effect, which could take place during the sample storage.

L161-162: "The wide ranges over about 15‰ in $\delta^{18}O$ and about 124‰ in $\delta^2H$ of monthly means of the new Inuvik data set enable a rather well-defined LMWL." The wide ranges in $\delta^{18}O$ and $\delta^2H$ values are also characteristic for the old dataset as well.

Anyway, in L138 "dataset" is written, while in other places "data set" (e.g. in L100). The same form should be used all over the paper.

---

## Author Comment (AC1)

**RC1: 'Comment on essd-2021-294', Trevor Porter, 08 Oct 2021**

**General comments**

The preprint is well written, clear and concise. It covers all the points I feel are necessary to describe this dataset. Scientifically, it is an important dataset and ESSD is the correct forum to present this record. Fritz et al. introduce a new precipitation isotope dataset from Inuvik, NT Canada, developed from 134 precipitation event samples collected from August 2015 to August 2018. The new sample size dwarfs the previous record available for this area (e.g., the original Inuvik record was defined by 14 monthly observations from the late 1980s). More broadly, the network of precipitation isotope records in the Canadian Arctic is very sparse in space and time, and badly out of date. Despite these limitations/uncertainties, Arctic precipitation isotope data have played critical roles in guiding the interpretation of environmental isotope datasets across a broad spectrum of research disciplines in paleoclimate (e.g., Porter et al., 2019, Nature Comm.; Holland et al., 2020, Geophys. Res. Lett.), cryosphere (Mackay, 1990, Can. J. Earth Sci.; Lacelle et al., 2014, Chem. Geol; Bandara et al. 2020, Permafr. Periglac. Process.) and hydrological sciences (Turner et al., 2014, Glob. Change Biol), and many others. However, the existing high-latitude precipitation isotope network including key stations such as Inuvik is now badly out of date, and the extent to which these old data are representative of modern precipitation-isotope systematics *was* (before this study) unclear after so many decades of change in climate and boundary conditions. Inuvik is major research center in the Canadian Arctic that invites hundreds of researchers from around the world each summer. A major update to the Inuvik isotope record was long overdue. This contribution will help advance our knowledge of local to regional precipitation-isotope systematics, which, in turn, will benefit an international and interdisciplinary community of researchers who are focused on the NW Canadian Arctic. I look forward to seeing this paper published in ESSD. I have only minor comments for improvement listed below.

**AR:** Thank you for your overall positive reception of our study. We are grateful for your comments whose implementation improves the presentation of the data set. Our replies to your comments are outlined in detail below.

**Specific comments/questions**

**L40.** What is meant by 'tackled by'? Clarify in simpler terms.

**AR:** Changed accordingly to: 'A LMWL can deviate from the GMWL both in slope and intercept of the linear regression in a δ18O-δ2H co-isotope plot. Such deviation largely results from differences in humidity (e.g. Putman et al., 2019) that is represented by the second-order parameter Deuterium excess (*d*; Dansgaard, 1964). Deuterium excess is expressed by the equation (2): …'

**L59-61.** This is a long sentence. Break it up if possible. I would also recommend elaborating on the significance of the Inuvik station. Inuvik is one of the most important centers of research in the NW Canadian Arctic.

**AR:** Changed accordingly in the revised ms: 'The present data collection aims (1) to update and extend the previous Inuvik LMWL and, thus, (2) to improve the regional framework for meteorological, hydrological and paleoclimate applications of precipitation stable isotopes of modern and past environments. Special emphasis is given to the central role of Inuvik for research in the western Canadian Arctic.'

**L86-88.** It seems very reasonable to exclude only one sample, but you should still explain *why* it should be excluded. Describing it as 'unusual' is ambiguous. Please be more descriptive.

**AR:** We added the following paragraph into the manuscript: "We consider two possible explanations which are (1) a wrongly labelled date on the sample bottle or (2) sublimation and according kinetic fractionation of a snow sample. As the February 2016 sample is the only one in the entire sample set showing such anomalous values, while it was stored and processed as all other samples, we decided

to note its values for completeness, but to exclude it from interpretation due to the untraceable origin of this erroneous value."

→ See also reply to Rev#2.

**L96.** Table caption, suggest revising to: Summary of precipitation events by month and year.

**AR:** Changed accordingly.

**L99.** "During this 4-year period only 14 monthly observations of d2H and d18O are available." Or something like this. Current sentence is a bit confusing.

**AR:** Changed accordingly.

**L135-136.** Porter et al. (2016, QSR – see Fig. 3b) also demonstrate higher d-excess during winter (colder) months across the broader Arctic GNIP network.

**AR:** Reference added.

**L141.** Can you elaborate? Also, I do understand that whether or not the old vs. new LMWL is not very important. This is an important data contribution. However, since you are comparing old vs. new LMWL's, you might consider a t-test based on +/- 1 sigma of the slope and intercept.

**AR:** The table below shows results of unpaired t-tests for different parameters between new and old Inuvik LMWL data sets. Critical t value was 1.6794. The results show that the null-hypothesis needs to be rejected, which means that the new and the old LMWL from Inuvik are statistically different from each other. We have inserted the according information into the revised version of the ms.

| δ18O | t(45) = 4.23, p < 0.001 |
|---|---|
| δ2H | t(45) = 3.80, p < 0.001 |
| Deuterium excess | t(45) = -5.16, p < 0.00001 |
| Slope | t(45) = 3.76, p < 0.001 |
| Intercept | t(45) = 5.16, p < 0.00001 |

**L144-145.** In this caption, add the n=34 and n=14 inside the brackets with the years.

**AR:** Changed accordingly.

**L156.** See comment about L141. Is this difference significant? If the difference is significant, probably the most important reason is that the old LMWL is defined by very few data points and may not be so representative. I think the following discussion in this paragraph is good, but you might try to simplify the sentences a bit. And please add a clear statement that the new LMWL is likely a better characterisation of local precipitation isotope dynamics that the old LMWL, simply on the basis of greater sample size.

**AR:** This reviewer comment relates to the deviation of the new Inuvik LMWL slope compared to the GMWL slope. We think that significance testing of the slope between the new Inuvik LMWL and the GMWL is not warranted, because the data pairs for the GMWL (Craig, 1961) are not publically available. Furthermore, we have already calculated the significant difference between old and new LMWL. As these are more similar to each other than to the GMWL a "significant" difference between new LMWL and GMWL can assumed. Nevertheless, we did not and we do not mention any statistical significant difference between the two.

We re-structured this paragraph and made shorter sentences to improve its readability. We furthermore added a statement on the greater sample size on which the new Inuvik LMWL is based on to the sentence as follows in the revised ms: 'The wide ranges over about 15‰ in $\delta^{18}O$ and about 124‰ in $\delta^{2}H$ of monthly means of the new Inuvik data set and the substantially greater sample size if compared to the old Inuvik LMWL enable a well-defined new LMWL.'

**L158-161.** Sentence is too long. Please break it up into smaller, more readable sentences.

**AR:** Changed accordingly.

**Technical corrections**

**L23.** Use proper multiplication symbol.

**AR:** Changed here and elsewhere in the ms accordingly.

**L39.** Improper use of semi-colon. Please revise the following sentences.

**AR:** Changed accordingly in the revised ms to: 'A LMWL can deviate from the GMWL both in slope and intercept of the linear regression in a $\delta^{18}O$-$\delta^2H$ co-isotope plot. Such deviation largely results from differences in humidity (e.g. Putman et al., 2019) that is represented by the second-order parameter Deuterium excess (*d*; Dansgaard, 1964). Deuterium excess is expressed by the equation (2): ...'

**RC2: 'Comment on essd-2021-294', István Fórizs, 15 Oct 2021**

**General comments**

The preprint is exceptionally well written. Very good English and concise wording. The whole preprint reflects that the authors have thorough knowledge of the given field of science. Plus, I could not identify any mistakes or faults in the referencing, which is rare.

**AR:** Thank you for reviewing our ms. We appreciate your suggestions to explain better the potential impact of sample storage on isotopic compositions. Please, find our detailed replies below and the according changes and additions in the revised ms.

**Specific comments/questions**

The authors goal is to provide an up-to-date and more robust dataset for calculating the local meteoric water line (LMWL) at Inuvik, Canada. To accomplish this goal, they collected event based precipitation samples from August 2015 to August 2018. The samples were stored in LDPE (low-density polyethylene) bottles at 4 °C. At this point I have a concern: the authors don't mention how long samples were stored before analysis. Samples were stored in Canada and the analyses were made in Germany, far away from each other, so it is reasonable to suppose that the samples were stored for several months. The International Atomic Energy Agency recommends HDPE (high-density polyethylene) bottles for storing water samples for several months, because LDPE bottles are not reliable. This is a very important question to make sure that the stored water samples have not suffered evaporation effect, because there is a small difference between the old and the new dataset and the most significant difference is in the d-excess. The average d-excess of the new dataset is significantly lower (5.7‰) than that of the old dataset (14.9‰). This difference can be caused by a) evaporation during sample storage, or b) climate change. Therefore, it is of paramount importance to check the reliability of sample storage. I recommend authors to discuss this question in details. If they cannot rule out the possibility of the evaporation effect during the sample storage, then their results are very ambiguous. If they can, then the manuscript is practically ready for publication. I recommend to check the sample storage related to the old dataset as well.

**AR:** Rev#2 is right raising the issues of storage time between sampling and isotopic analysis and the type of the sample container. Indeed, borosilicate glass bottles or HDPE bottles hold the lowest chance for fractionation processes during long storage periods. However, a current study by Spangenberg (2012) comparing the control of different storage container materials on isotopic water composition after 659 days of storage notes the following:

(1) No clear trend in the $\delta^2$H and $\delta^{18}$O variations for water stored in the organic polymer bottles such as HDPE or LDPE bottles.

(2) The differences in the $\delta^2$H-$\delta^{18}$O covariations of LDPE containers are best explained by differences in container wall thickness and volume which are both in our study the same for all samples.

In the storage test study by Spangenberg (2012), the potential isotopic variations in LDPE bottles (10 mL and 50 mL), which are close to the LDPE sample bottles used in our study (15 mL, 30 mL), show no trend over time and range from about 0.5 to 1.0 ‰ in $\delta^{18}$O and from about 1.9 to 3.0 ‰ in $\delta^2$H. Given the only minor effects of the LDPE material on the isotopic composition on monthly time scales, the container material we used (Kautex narrow-neck LDPE) provides some practical advantages for sampling in high latitudes. Those are (1) that the material is flexible to ensure complete filling without remaining air in the headspace, (2) that the material does not break upon freezing or mechanical stress, and (3) complete and very tight closure of the lid to avoid exchange with ambient air and sample loss due to evaporation. A previous long-term testing (analyses after 6 and 12 months) of the LDPE bottles that we used, revealed identical values within the analytical error range of our mass spec. Therefore, we are confident that neither evaporation nor other substantial fractionation processes altered the isotopic composition significantly.

We can further ensure a maximum period between sampling and analysis of <12 months. This time period is defined between the sampling on site in Inuvik and the sample pick-up and transport every summer by scientists from the AWI in Potsdam, who carry out fieldwork out of Inuvik every year. This means we never reached the storage duration mentioned in the experimental study by Spangenberg (2012), which was 659 days.

This information was added to the Material and Methods chapter in the ms.

**Mostly technical corrections**

Abbreviations (WMO in L65; AWI in Caption of Fig. 1; ARI in L81) should be defined at the first instance.

**AR:** Changed accordingly.

**L88:** The d value is -17.8‰, and not 17.8‰. Anyway, this very negative d value indicates evaporation effect, which could take place during the sample storage.

**AR:** Rev#2 is right, the deuterium excess values of the excluded sample from February 2016 amounts indeed to –17.8‰, but got wrongly typed in the submitted ms. Thank you for noting. All available winter samples of our data set considering the months November to March never show negative deuterium excess values. Especially, the only comparable February data (2017, n=5) varies from 6.5 to 13.5‰ in deuterium excess putting the –17.8‰ value from Feb 2016 certainly under question. We consider two possible explanations which are (1) a wrongly labelled date on the sample bottle or (2) sublimation and according kinetic fractionation of a snow sample. As the Feb 2016 sample is the only one in the entire sample set showing anomalous values, while it was stored and processed as all other samples, we decided to note its values in the ms for completeness, but to exclude it from interpretation due to the untraceable origin of this erroneous value. This paragraph was added into the revised ms.

→ See also Rev#1.

We exclude substantial alternation of the isotopic composition by evaporation due to long-term testing of our sample bottles. See above.

**L161-162:** "The wide ranges over about 15‰ in δ18O and about 124‰ in δ2H of monthly means of the new Inuvik data set enable a rather well-defined LMWL." The wide ranges in δ18O and δ2H values are also characteristic for the old dataset as well.

**AR:** Agreed. As also recommended by rev#1, we added a statement on the greater sample size on which the new Inuvik LMWL is based on to the sentence as follows in the revised ms: 'The wide ranges over about 15‰ in $\delta^{18}$O and about 124‰ in $\delta^2$H of monthly means of the new Inuvik data set and the substantially greater sample size if compared to the old Inuvik LMWL enable a rather well-defined new LMWL.'

Anyway, in **L138** "dataset" is written, while in other places "data set" (e.g. in L100). The same form should be used all over the paper.

**AR:** Changed accordingly throughout the paper to 'data set'.

**RC3: 'Comment on essd-2021-294', Anonymous Referee #3, 26 Oct 2021**

This data description paper by Fritz and co-authors presents a very useful data set which covers a data gap in high latitude precipitation at Inuvik over recent periods. Having isotopic data from high latitude stations is always a difficult task but of paramount importance when we look at past and recent climate changes in the Arctic. The paper accompanying the data is well written and well structured.

I have only few minor comments/questions regarding the quite negative deuterium excess values, the sampling procedure, and the way the mean monthly values are calculated. I would like the authors to comments on these points.

AR: We appreciate your overall positive feedback on our study and thank you for your time and effort to review the ms. Our replies to your points raised are outlined in detail below.

Negative d excess values: I noticed in the data sets that quite negative values are present, mostly (BUT not only …) centred during summer months. Can you exclude evaporation effects? Are those samples properly preserved/collected?

AR: We certainly assume proper collection and preservation of the samples. Since a similar concern was expressed by rev#2, we added related information on sample storage duration and sample container material in the Material and Methods section of the revised ms.

→ see also our according reply to rev#2

As a side note: Negative summer deuterium excess values nowadays compared to the past can possibly be attributed to climate warming (higher air temperatures, higher evaporation, or contribution of recycled (secondary) moisture, e.g. Bonne et al. 2020). But this might be part of further data analysis and discussion in the future, which we want to promote by publishing the current data set.

Regarding the sampling procedure: how did you perform the snow sampling? Usually collecting snow, it is not an easy task, particularly using the "normal rain" collectors. It was not clear in the text.

AR: Correct. Collecting snow is not an easy task, particularly using the "normal rain" collectors. Snow was scraped out from the funnel of the precipitation gauge after a snow event. The snow was placed into a larger (250mL) bottle, closed and let melt. Then the meltwater was poured into the standard sample bottle. We added a sentence on snow sampling to the ms.

How did you calculate the mean monthly values? Is this an arithmetical mean or you averaged out the values considering the precipitation amount? This is quite important since you then use the monthly data for the LMWL. Adding the precipitation amounts, if available, could be an interesting information, although I can understand that this request could be not satisfied.

AR: As we did not measure the event-based precipitation amount, but collected solely about 30 mL per event for further analysis, we calculated the arithmetic mean.

May you add in Table 3 the elevation of the different stations?

AR: Changed accordingly.

**References:**

Bonne, J.-L., Meyer, H., Behrens, M., Boike, J., Kipfstuhl, S., Rabe, B., Schmidt, T., Schönicke, L., Steen-Larsen, H. C., and Werner, M.: Moisture origin as a driver of temporal variabilities of the water vapour isotopic composition in the Lena River Delta, Siberia, Atmos. Chem. Phys., 20, 10493–10511, https://doi.org/10.5194/acp-20-10493-2020, 2020.

Porter, T.J., Froese, D.G., Feakins, S.J., Bindeman, I.N., Mahony, M.E., Pautler, B.G., Reichart, G.-J., Sanborn, P.T., Simpson, M.J., and Weijers, J.W.H.: Multiple water isotope proxy reconstruction of extremely low last glacial temperatures in Eastern Beringia (Western Arctic), Quat. Sci. Rev., 137, 113–125, https://doi.org/10.1016/j.quascirev.2016.02.006, 2016.

Spangenberg, J.E.: Caution on the storage of waters and aqueous solutions in plastic containers for hydrogen and oxygen stable isotope analysis, Rapid Communications in Mass Spectrometry, 26, 2627-2636, https://doi.org/10.1002/rcm.6386, 2012.